# Ag_3_PO_4_ Deposited on CuBi_2_O_4_ to Construct Z-Scheme Photocatalyst with Excellent Visible-Light Catalytic Performance Toward the Degradation of Diclofenac Sodium

**DOI:** 10.3390/nano9070959

**Published:** 2019-06-30

**Authors:** Xiaojuan Chen, Chunmu Yu, Runliang Zhu, Ning Li, Jieming Chen, Shuai Li, Wei Xia, Song Xu, Hailong Wang, Xin Chen

**Affiliations:** 1College of Environmental and Chemical Engineering, Foshan University, Foshan 528000, China; 2CAS Key Laboratory of Mineralogy and Metallogeny, Guangdong Provincial Key Laboratory of Mineral Physics and Material, Guangzhou Institute of Geochemistry, Chinese Academy of Sciences, Guangzhou 510640, China; 3CAS Key Laboratory of Renewable Energy, Guangzhou Institute of Energy Conversion, Chinese Academy of Sciences, Guangzhou 510650, China; 4College of Transportation and Civil Architecture, Foshan University, Foshan 528225, China

**Keywords:** Z-Scheme, CuBi_2_O_4_/Ag_3_PO_4_, visible light, photocatalytic mechanism

## Abstract

CuBi_2_O_4_/Ag_3_PO_4_ was synthesized through a combination of hydrothermal synthesis and an in situ deposition method with sodium stearate as additives, and their textures were characterized with XRD, XPS, SEM/HRTEM, EDS, UV-Vis, and PL. Then, the photodegradation performance of CuBi_2_O_4_/Ag_3_PO_4_ toward the degradation of diclofenac sodium (DS) was investigated, and the results indicate that the degradation rate of DS in a CuBi_2_O_4_/Ag_3_PO_4_ (1:1) system is 0.0143 min^−1^, which is 3.6 times that in the blank irradiation system. Finally, the photocatalytic mechanism of CuBi_2_O_4_/Ag_3_PO_4_ was discussed, which follows the Z-Scheme theory, and the performance enhancement of CuBi_2_O_4_/Ag_3_PO_4_ was attributed to the improved separation efficiency of photogenerated electron–hole pairs.

## 1. Introduction

Diclofenac sodium (DS) is one of the most widely used nonsteroidal anti-inflammatory drugs [1,2]. However, most DS will be excreted outside of the body through urine and stool. Since the removal rate of DS after the treatment of conventional wastewater treatment plants (WWTPs) is no more than 20%, most of the remaining DS would be discharged along with effluents into estuaries, rivers, surface water, ground water, and even drinking water [3,4,5]. The DS frequently detected in aquatic environments (estuaries, rivers, surface water, ground water, and even drinking water) has aroused great concern in that it endangers human health and poses great risks to the environment [6,7]. Over the last several years, various technologies encompassing adsorption [7,8], photocatalysis [9], Fenton reagents [10], ozonation [11], etc. have been studied to remove DS. Semiconductor photocatalysis is considered a green environmentally friendly and cost-efficient technology [12,13]. However, many of the known photocatalysts such as TiO_2_ [14,15], CeO_2_ [16,17], AgCl [18], ZnO [19], etc. have large energy band gaps, making them able to only utilize ultraviolet (UV) light to stimulate their activity. Unfortunately, the region of UV only occupies approximately 4% of the entire solar spectrum, which is just 9.3% of visible light. Considering the above, great efforts have been put in the development of visible-light responsive photocatalysts in recent years [12,13,20].

Silver orthophosphate (Ag_3_PO_4_), a visible-light responsive photocatalyst with a bandgap of 2.36–2.43 eV, has aroused great attraction since its discovery. [21]. As reported, in the photoelectrocatalysis system of Ag_3_PO_4_ for water oxidation with a radiation wavelength around 420 nm, a quantum efficiency of approximately 90% could be achieved, which is apparently higher than the referenced semiconductors of approximately 20% (e.g., BiVO_4_ or N-doped TiO_2_) [21]. However, this high efficiency occurs in a system with AgNO_3_ as an electron scavenger. If the scavenger does not exist, the activity of Ag_3_PO_4_ would be seriously decreased due to photocorrosion. The photocorrosion of Ag_3_PO_4_ is closely related to the slight solubility (0.02 g/L) and the reduction of Ag^+^ to Ag^0^ (4Ag_3_PO_4_ + 6H_2_O + 12*h*^+^ + 12*e*^−^ → 12Ag^0^ + 4H_3_PO_4_ + 3O_2_). The above analysis indicated that there still exists a great challenge in maintaining the stability of Ag_3_PO_4_ in the absence of electron scavengers.

Recently, numerous strategies, including morphology control [22,23], ion doping [24,25], coupling with other semiconductor to construct heterojunction or Z-Scheme composite photocatalysts [26,27,28], combining with support materials [29,30,31], etc., have tried to enhance the photocorrosion resistance of Ag_3_PO_4_. Z-scheme photocatalytic, with a wide absorption range, high charge-separation efficiency, and strong redox ability, not only can overcome the drawbacks of single-component photocatalysts and possesses long-term stability but also can be used to mimic the natural photosynthesis process. Therefore, the construction of Z-Scheme composite photocatalysts has attracted great attention. In order to construct the Z-Scheme composite photocatalysts, appropriate semiconductors of energy band structures matched with Ag_3_PO_4_ are necessary. CuBi_2_O_4_ is a novel photocatalytic material with a strong visible-light response and excellent photostability [32]. However, previous studies demonstrated that CuBi_2_O_4_ exhibited poor performance on its own, and its composite via doping with other metal oxides semiconductor shows higher activity [33]. Because of the conduction band (CB) potential and valence band (VB) potential of Ag_3_PO_4_ being more positive than that of CuBi_2_O_4_, theoretically speaking, it is feasible to combine Ag_3_PO_4_ and CuBi_2_O_4_ to form a Z-Scheme composite photocatalyst.

Thus, in this study, the well-distributed spherical CuBi_2_O_4_ was first prepared under different conditions through the hydrothermal synthesis method and, then, the composite CuBi_2_O_4_/Ag_3_PO_4_ was further synthesized by the in situ deposition method with sodium stearate as additives. The photocatalytic performances of these catalysts were evaluated for DS photodegradation under visible-light irradiation. Moreover, the photocatalytic mechanism of the catalysts was investigated in detail.

## 2. Materials and Methods 

### 2.1. Synthesis of CuBi_2_O_4_/Ag_3_PO_4_

#### 2.1.1. Synthesis of CuBi_2_O_4_

CuBi_2_O_4_ with different morphologies were synthesized with the hydrothermal method, which includes four steps. Firstly, Bi(NO_3_)_3_·5H_2_O and Cu(NO_3_)_2_·3H_2_O were dissolved into moderate volumes of HNO_3_ and HNO_3_ was prepared to dissolve the Bi(NO_3_)_3_·5H_2_O precursor; secondly, the precipitator of NaOH was added into the reaction system. Thirdly, the mixture solution was diluted to a certain volume; fourth, the solution was reacted hydrothermally under 100 °C for 24 h. After the isolation of the produced precipitates by centrifugation, drying of these solids at 60 °C overnight is necessary.

#### 2.1.2. Synthesis of CuBi_2_O_4_/Ag_3_PO_4_

The in situ deposition method was used to prepare CuBi_2_O_4_/Ag_3_PO_4_ with sodium stearate as additives. The detailed preparation processes of CuBi_2_O_4_/Ag_3_PO_4_ for the mass ratio of 1:1 is as follows: 0.1256 g CuBi_2_O_4_ was dispersed into 40 mL of ultrapure water, the solution was ultrasound at 100 W for 15 min subsequently, and then a sodium stearate solution (10 mL, the mole ratio of the sodium stearate to Ag^+^ is 2) was added into the reaction system and mechanically agitated for 2 h. After that, an AgNO_3_ solution (10 mL, 0.9 mmol) was added to the mixture. After stirring for 30 min, a Na_2_HPO_4_·12H_2_O solution (20 mL, 0.3 mmol) was added drop by drop. The precipitate was isolated and washed several times with absolute ethanol and distilled water and then dried at 60 °C overnight.

### 2.2. Characterization of Photocatalysts

The morphologies of as-prepared catalysts were characterized with JSM-6360LV (JEOL, Tokyo, Japan) and high-resolution transmission electron microscopy (HRTEM, FEI Tecnai G^2^ F20 S-TWIN, Hillsboro, OR, USA). The crystal structures of various catalysts were examined with the X-ray diffraction instrument (XRD, Rigaku D/max 2500 PC, Rigaku, Japan). The surface element composition and chemical state analysis were performed on the X-ray photoelectron spectroscopy (XPS) spectra (ESCALAB 250 spectrometer, Waltham, MA, USA). Energy-dispersive X-ray spectroscopy (EDS) was also used to investigate the types and contents of elements in the materials. The nitrogen adsorption–desorption isotherms were obtained using a NOVA-2200e volumetric analyzer (Quantachrome, Boynton Beach, FL, USA). The surface areas of the samples can also be estimated by the BET model. The optical absorption performance of the photocatalysts was analyzed with UV-vis diffusive reflectance spectra labeled by Shimadzu UV-2550 (Kyoto, Japan). The photoluminescence (PL) spectroscopy was detected with a JASCO, FP-6500 florescence spectrophotometer (Oklahoma City, OK, USA).

### 2.3. Experiments of Photocatalytic Performance

Photocatalytic activity experiments: The photocatalytic activity of as-prepared catalysts was evaluated toward the degradation of diclofenac sodium (DS) as well as dyestuff of Rhodamine B (RhB), Congo red (CR), Methyl Orange (MO), and Methylene Blue (MB). The photoreaction apparatus, equipped with a 300 W Xe lamp and an ultraviolet cutoff filter providing visible light at ≥400 nm, was manufactured by Bilang Biological Science and Technology Co., Ltd., Xi’an, China. The configuration of the apparatus is illustrated in our previous paper [34]. During the experiments, 0.025 g of photocatalyst was added into a 50 mL 10 mg/L DS solution. After the preparation, the adsorption–desorption equilibrium experiment was firstly onset in the dark with magnetically stirred for 30 min. Then, the Xe lamp light was turned on, and the sample was withdrawn at a setting time interval and was filtered with 0.45-µm membrane filters. The pH, concentration of residual DS, and TOC in the filter were detected with the pH analyzer (PHS-3C, Shanghai Jinghong Scientific Instrument Co. Ltd., Shanghai, China), high-performance liquid chromatography (HPLC, Agilent 1260, Santa Clara, CA, USA), and TOC (Shimadzu TOC-V_CPH,_ Kyoto, Japan) analyzer, respectively. As for the photodegradation of dyestuff in the CuBi_2_O_4_/Ag_3_PO_4_ (1:1) system, the concentration of dyestuff is 15 mg/L and the catalyst dosage is 0.5 g/L.

Photocatalytic stability experiments: The stability of CuBi_2_O_4_/Ag_3_PO_4_ (1:1) was demonstrated with a repeatability test. The specific experimental processes can refer to the photocatalytic activity experiments. After the reaction in each run, the photocatalysts were collected and vacuum-dried overnight at 60 °C for recycling use.

### 2.4. Analysis of Reactive Species

The reactive species formed during the photodegradation process of DS were checked with free radical capture experiments, and tert-butanol (t-BuOH), disodium ethylenediamine tetraacetate (EDTA-Na_2_), benzoquinone (BZQ) were chose as the scavenger of hydroxyl radical (OH), the hole (*h*^+^), and the superoxide radical (O_2_^−^), respectively. The detailed free radical capture experiment processes were similar to the photocatalytic activity experiments.

## 3. Results and Discussion

### 3.1. Morphology and Associated Formation Mechanism of CuBi_2_O_4_

Four different experiments under various conditions for the preparation of CuBi_2_O_4_ were firstly conducted to investigate the effect of the morphology and structure on the photocatalytic activity. SEM images of the as-prepared CuBi_2_O_4_ are shown in Figure 1a–d. As shown in Figure 1a, the CuBi_2_O_4_ prepared with 1.2 M NaOH and the step of “dilution” after the addition of NaOH displays well-distributed microsphere structures with diameters of ~4 μm, and its surfaces are comprised of nanometer quadrilateral plates with lengths of ~250 nm and widths of ~200 nm. When the mixture solution of Bi(NO_3_)_3_·5H_2_O + Cu(NO_3_)_2_·3H_2_O + HNO_3_ was diluted before the addition of 1.2 M NaOH, the CuBi_2_O_4_ with a relatively compact and smooth surface are obtained, as described in Figure 1b. Figure 1c,d shows the SEM images of CuBi_2_O_4_ prepared with 2.4 M NaOH, and the step of “dilution” was after or before the addition of NaOH, respectively. The microsphere structural CuBi_2_O_4_ with its surface comprised of tens of nanometers cube was obtained when the 2.4 M NaOH was added before the “dilution”, and the CuBi_2_O_4_ with nonuniform size and structure were obtained when the 2.4 M NaOH was added after the “dilution”. The remarkable influence of NaOH molarity on the morphology and property of CuBi_2_O_4_ was also evidenced by others. Wang et al. [35] reported the hedgehog-like CuBi_2_O_4_ hierarchical microspheres. Oh et al. [36] reported the three-dimensional spherical CuBi_2_O_4_ nan column arrays connected through non-covalent interactions. Patil et al. [37] reported the spherulitic flower morphology of CuBi_2_O_4_. Xie et al. [38] reported the leaf-like structural CuBi_2_O_4_. Moreover, the order of “dilution” mainly influences the mass transfer rate and Gibbs free energy of the reaction system, finally leading to the formation of different structural crystals to achieve the lowest Gibbs free energy [36].

On the whole, the mechanism of CuBi_2_O_4_ formation involves three consecutive stages: First, the dissolution of Bi(NO_3_)_3_·5H_2_O in HNO_3_, as described by the Equation (1); second, the precipitation of Bi(OH)_3_ and Cu(OH)_2_ because the solubility product constants of Bi(OH)_3_ and Cu(OH)_2_ are 2.9 × 10^−7^ and 4.8 × 10^−20^ (25 °C), respectively, so when the precipitating agent of NaOH was added into the reaction system, the blue flocculent precipitates of Cu(OH)_2_ formed first, followed by the formation of white precipitates Bi(OH)_3_, which can be described by Equations (2) and (3); and third, the formation and Ostwald-ripening process of CuBi_2_O_4_, in which the Bi(OH)_3_ would lose one water molecule first to become partial yellow bismuth hydroxide BiO(OH) under 100 °C (see Equation (4)) and reacted with Cu(OH)_2_ to form CuBi_2_O_4_ (see Equation (5)) and then the as-formed CuBi_2_O_4_ crystals gave birth to different morphologies via Ostwald ripening.
(1)$$\mathrm{Bi}{\left({\mathrm{NO}}_{3}\right)}_{3}\cdot 5H_{2}O\overset{{\mathrm{HNO}}_{3}}{\to }Bi^{3+}+3{\mathrm{NO}}_{3}{}^{-}+5H_{2}O$$
(2)$$Cu^{2+}+2OH^{-}\to \mathrm{Cu}{\left(\mathrm{OH}\right)}_{2}↓$$
(3)$$Bi^{3+}+3OH^{-}\to \mathrm{Bi}{\left(\mathrm{OH}\right)}_{3}\;↓$$
(4)$$\mathrm{Bi}{\left(\mathrm{OH}\right)}_{3}\overset{100\;℃}{\to }\mathrm{BiO}\left(\mathrm{OH}\right)+H_{2}O$$
(5)$$2\mathrm{BiO}\left(\mathrm{OH}\right)+\mathrm{Cu}{\left(\mathrm{OH}\right)}_{2}\overset{100\;℃}{\to }{\mathrm{CuBi}}_{2}O_{4}↓\;+2H_{2}O$$

The products of CuBi_2_O_4_ with different morphologies shown in Figure 1a–d were recorded as “Product a”, “Product b”, “Product c”, and “Product d”, respectively. Their photocatalytic activity toward DS degradation are described in Figure 1e. As can be seen, all the CuBi_2_O_4_ exhibited more enhanced photocatalytic activity than the self-photodegradation of DS and “Product a” showed the highest catalytic activity, which is consistent with the variation of specific surface area (see Figure 1f). Therefore, “Product a” was used to further synthesize the composite CuBi_2_O_4_/Ag_3_PO_4_.

### 3.2. Characterization of CuBi_2_O_4_/Ag_3_PO_4_

#### 3.2.1. Morphology

The morphology of the as-prepared catalysts are characterized by SEM, and the results are shown in Figure 2. Figure 2a shows the SEM image of Ag_3_PO_4_, which exhibits an irregular polyhedron structure with an average diameter of approximately 400 nm. Compared with the morphology of CuBi_2_O_4_ shown in Figure 1a, the surface of composite CuBi_2_O_4_/Ag_3_PO_4_ (1:1) becomes rough (see Figure 2b), which is because the nano-particulate Ag_3_PO_4_ are attached onto the surface and occupies the nano-sheet gap of the CuBi_2_O_4_′s surface. However, the size of Ag_3_PO_4_ in the composite is tailored to be approximately ~100 nm, mainly because of the inhibition of the nano-sheet gap of the CuBi_2_O_4_′s surface on the nucleation, growth, and crystallization processes of Ag_3_PO_4_. Figure 2c shows the lattice fringe image of CuBi_2_O_4_/Ag_3_PO_4_ (1:1). The observed lattice fringes of 0.267 nm and 0.240 nm correspond to the (210) and (211) planes of Ag_3_PO_4_, while the *d* spacings of 0.320 nm, 0.426 nm, 0.292 nm, and 0.271 nm can be assigned to the (211), (200), (002), and (310) planes of CuBi_2_O_4_. The composites of prepared CuBi_2_O_4_/Ag_3_PO_4_ (1:1) were also analyzed by the elemental signature from the EDS spectrum (Figure 2d). The spectrum suggests that the composites are composed of O, Bi, Cu, Ag, and P and that their proportion are 50.34%, 10.30%, 3.10%, 27.47%, and 8.79%, respectively, which further indicates the component of CuBi_2_O_4_/Ag_3_PO_4_.

#### 3.2.2. Component and Surface Property

The crystallinity and component of the as-prepared materials were characterized by XRD, and the results are shown in Figure 3. It can be clearly seen that the diffraction peaks in the XRD pattern of Ag_3_PO_4_ can be indexed to the phase of Ag_3_PO_4_ (JCPDS No. 06-0505) [39]. Moreover, the reflections at 2*θ* = 20.76, 29.58, 33.20, 36.48, 42.38, 47.75, 52.61, 54.94, 57.19, 61.56, 65.65, 69.80,71.79, 87.19, and 89.02 are attributed to the crystal planes of (110), (200), (210), (211), (220), (310), (222), (320), (321), (400), (330), (420), (421), (432), and (521), respectively. From the XRD pattern of CuBi_2_O_4_, the diffraction pattern shows peaks at 2*θ* = 20.46, 27.67, 29.21, 30.80, 32.66, 34.25, 37.29, 46.07, 55.05, 59.75, and 65.88, which are in good agreement with the diffractions from (200), (211), (220), (002), (102), (310), (202), (411), (332), (521), and (413), respectively (JCPDS No. 84-1969) [40]. Furthermore, the XRD pattern of CuBi_2_O_4_/Ag_3_PO_4_ (1:1) suggests that the material obtained just contains CuBi_2_O_4_ and Ag_3_PO_4_, with the absence of any other substances, indicating that Ag_3_PO_4_ couples with CuBi_2_O_4_ mainly through the physical effects but not the chemical reaction.

The chemical compositions and surface chemical states of the CuBi_2_O_4_/Ag_3_PO_4_ (1:1) composite photocatalyst were further confirmed by the X-ray photoelectron spectroscopy (XPS), as shown in Figure 4. The survey spectrum of CuBi_2_O_4_/Ag_3_PO_4_ (1:1), shown in Figure 4a, indicates the presence of Cu, Bi, Ag, P, and O. Figure 4b presents the Cu 2p region. It can be clearly observed that the two main peaks at 934.25 eV and 954.18 eV with a spin-orbit splitting of 20.13 eV are assigned to the binding energies of Cu 2p_3/2_ and Cu 2p_1/2_, and the two satellite peaks observed at 942.07 eV and 963.25 eV further confirm the Cu^2+^ valence state [41]. Figure 4c shows the core level of the Bi 4f spectra; two bands at 159.13 eV and 164.44 eV corresponding to the binding energies of Bi 4f_7/2_ and Bi 4f_5/2_, respectively, are observed. The results are consistent with the studies reported by other researchers [36]. Figure 4d shows the Ag 3d region, including two highly intense peaks at 368.16 eV and 374.17 eV, which can be ascribed to the Ag 3d_5/2_ and Ag 3d_3/2_ binding energies [42,43], respectively. For the high-resolution XPS spectrum of P 2p, shown in Figure 4e, only one peak with the binding energy of 132.90 eV is observed, and it is extremely similar to the 132.40 eV evolved in the PO_4_^3−^ [44]. In Figure 4f, the XPS spectrum of O 1s indicating the binding energy of 530.79 eV represents the O-metal bonds, which is in agreement with the reported values of O^2−^ anion coming from CuBi_2_O_4_ and Ag_3_PO_4_ [41,44].

#### 3.2.3. Optical Absorption Property

The optical absorption properties of the as-prepared catalysts were determined by the UV-Vis diffusive reflectance analyzer, and the spectra are shown in Figure 5a. Both of the pure CuBi_2_O_4_ and Ag_3_PO_4_ exhibit strong absorbances in the UV and visible light regions, and their absorbance boundaries are 530 nm and >800 nm, respectively. The results are consistent with the former discovery by other teams [21,45]. In the case of the CuBi_2_O_4_/Ag_3_PO_4_ (1:1) composite, the absorbance in the visible light regions is much higher than that of the pure Ag_3_PO_4_. This property can make a positive contribution to the photocatalytic activity of the composite because a more efficient utilization of solar energy could be achieved [46]. In addition, according to the Kubelka–Munk function and the plot of (*αhv*)^2^ vs *hv* (shown in Figure 5b) [46,47], the band gaps (*E*g) of Ag_3_PO_4_, CuBi_2_O_4_, and CuBi_2_O_4_/Ag_3_PO_4_ (1:1) were estimated to be 2.42 eV, 1.72 eV, and 2.01 eV, respectively. The band gap energy of CuBi_2_O_4_/Ag_3_PO_4_ (1:1) is obviously narrower than that of bare Ag_3_PO_4_, which can conclude that the formed composite is more easily excited by visible light and that the utilization ratio of visible light is enhanced. Furthermore, the band-edge potentials of the conduction band (*E*_CB_) and valence band (*E*_VB_) could be calculated from the equations [47]: *E*_VB_ = *X* − *E*^C^ + 0.5*E*_g_ and *E*_CB_ = *X* − *E*^C^ − 0.5*E*_g_, where *X* is the geometric mean of the electronegativity of the constituent atoms (5.96 eV for Ag_3_PO_4_ and 4.59 eV for CuBi_2_O_4_) [26,48] and *E*^C^ is the energy of the free electrons on the hydrogen scale (about 4.5 eV). Thus, the *E*_VB_ and *E*_CB_ of Ag_3_PO_4_ can be estimated to be 2.67 eV/NHE and 0.25 eV/NHE, while the *E*_VB_ and *E*_CB_ of CuBi_2_O_4_ can be estimated to be 0.95 and −0.77 eV/NHE, respectively.

### 3.3. Photodegradation Performance of DS in Different Catalysts Systems

In order to estimate the photocatalytic activity of the as-prepared materials, DS was selected as a target pollutant. Figure 6a indicates the photodegradation efficiency of DS in different photocatalytic systems. In the blank irradiation system, the self-photodegradation efficiency of DS during the 120 min of reaction is 37.81%. Under the same conditions, the degradation efficiencies of DS in the pure CuBi_2_O_4_ and Ag_3_PO_4_ photocatalytic system are 67.12% and 77.84%, respectively. While in the photocatalytic system of CuBi_2_O_4_/Ag_3_PO_4_ (1:1), the degradation efficiency of DS reaches 85.45%. This suggests that the addition of catalysts promotes the degradation of DS and that there is a synergistic effect between CuBi_2_O_4_ and Ag_3_PO_4_. A pseudo-first-order kinetic model was employed to fit the experimental data of DS degradation in different photocatalytic systems, and the results are shown in Table 1. It can be seen that the degradation rate constants of DS are 0.0041 min^−1^, 0.0084 min^−1^, 0.0112 min^−1^, and 0.0069 min^−1^ in the photocatalytic system of blank irradiation, CuBi_2_O_4_, Ag_3_PO_4_, and commercial TiO_2_, respectively. The low photocatalytic activity of commercial TiO_2_ is mainly due to its weak absorption of visible light, resulting in a small number of active carriers. As for the photocatalytic systems of composite CuBi_2_O_4_/Ag_3_PO_4_, the rate constant of DS increases firstly and then decreases with the further increase of Ag_3_PO_4_ content in the composites, which is consistent with the variation of specific surface area (see Table 1). The main reason is that the deposition of excessive Ag_3_PO_4_ on the surface of CuBi_2_O_4_ will block the gap between the nanosheets on the surface of CuBi_2_O_4_, which is not conducive to the diffusion of reactants and products. Also, when the mass ratio of CuBi_2_O_4_ and Ag_3_PO_4_ in the composite is 1:1, the rate constant of 0.0143 min^−1^ can be obtained, which is approximately 3.6 times that of the values in the blank irradiation system.

To understand the mineralization rate of DS in different photocatalytic systems, the TOC concentrations of the reacted DS solutions were detected and the removal efficiency are described in Figure 6b. In the blank irradiation system, the mineralization rate of DS is 14.36%. When the catalysts of CuBi_2_O_4_, Ag_3_PO_4_, and CuBi_2_O_4_/Ag_3_PO_4_ (1:1) were introduced into the reaction system, the mineralization rates of DS increased to be 35.43%, 49.62%, and 57.33%, respectively. Figure 6c shows the change of the solution pH as reaction time goes on at different photocatalytic systems. In general, the pH of the solution decreases, but there is a fluctuation at a certain time interval in each photocatalytic system, which can be explained by forming some acidic intermediates, and they are decomposed as the reaction time is prolonged.

Figure 6d shows the removal efficiency of various dyes in the catalytic system of CuBi_2_O_4_/Ag_3_PO_4_ (1:1). It can be seen that, within 20 min, 15 mg/L of RhB and MB can be completely decomposed by 0.5 g/L of the composite photocatalyst, while the removal efficiency of CR and MO are 85.38% and 94.24%, respectively. The results show that the as-prepared CuBi_2_O_4_/Ag_3_PO_4_ exhibits excellent photocatalytic activity for the degradation of dyes as well.

The stability of the photocatalyst is a very important parameter with regard to practical applications. To further investigate the performance stability of the prepared CuBi_2_O_4_, Ag_3_PO_4_, and CuBi_2_O_4_/Ag_3_PO_4_ (1:1), the cycling degradation experiments of DS were carried out and the results are shown in Figure 7a. It is found that the degradation efficiency of DS in the pure CuBi_2_O_4_ photocatalytic system reduced from 67.12% to 63.05% after a 5-time repeated reaction, while the degradation efficiency of DS in the pure Ag_3_PO_4_ and CuBi_2_O_4_/Ag_3_PO_4_ (1:1) photocatalytic system reduced by a large extent, i.e., 18.51% and 11.99%. Furthermore, the corresponding XRD results shown in Figure 7b suggest that there is a negligible change about the phase structure of CuBi_2_O_4_ sample after the repeated photocatalytic reactions, further indicating the photocatalytic stability of the CuBi_2_O_4_. However, compared with the XRD spectra of the fresh Ag_3_PO_4_ and CuBi_2_O_4_/Ag_3_PO_4_ (1:1), the diffraction peaks readily indexed as the metallic silver (Ag^0^) emerge in the XRD spectra of the reacted Ag_3_PO_4_ and CuBi_2_O_4_/Ag_3_PO_4_ (1:1). In addition, the appearance of Ag^0^ in the reacted Ag_3_PO_4_ and CuBi_2_O_4_/Ag_3_PO_4_ (1:1) can be further confirmed by the XPS. Figure 7c shows the high-resolution Ag 3d XPS spectrum of CuBi_2_O_4_/Ag_3_PO_4_ (1:1) used five times. Based on the Ag 3d XPS spectrum of the fresh CuBi_2_O_4_/Ag_3_PO_4_ (1:1) (see Figure 4d), the two individual peaks are further divided into four different peaks and the two main peaks at binding energies of 368.16 eV and 374.17 eV can be attributed to the Ag^+^ of Ag_3_PO_4_, whereas the peaks at 368.32 eV and 374.40 eV can be attributed to the metallic silver (Ag^0^). From the result, it is clear that the Ag^0^ should have been formed on the surface of the reacted CuBi_2_O_4_/Ag_3_PO_4_ (1:1) and that the proportion of the peaks relative to Ag^0^ occupy 5.52% of the total fitting peak area. However, the Ag 3d XPS spectrum of the reacted Ag_3_PO_4_ shown in Figure 7d indicates that the formed Ag^0^ in the surface of Ag_3_PO_4_ is 11.78%, which is much higher than that in the reacted CuBi_2_O_4_/Ag_3_PO_4_ (1:1), further implying the enhanced stability of the CuBi_2_O_4_/Ag_3_PO_4_ (1:1) in comparison to pure Ag_3_PO_4_.

### 3.4. Photocatalytic Mechanism of Different Catalysts

Photoluminescence (PL) is a useful tool for obtaining information about the photogenerated electron–hole recombination property of materials, which helps us to understand the photocatalytic mechanism of the catalysts. Therefore, the PL spectra of the as-prepared CuBi_2_O_4_, Ag_3_PO_4_, and CuBi_2_O_4_/Ag_3_PO_4_ (1:1) were first recorded at room temperature with a excitation wavelength of 500 nm, and the spectra are shown in Figure 8a. Obviously, the PL intensity of the CuBi_2_O_4_/Ag_3_PO_4_ (1:1) composite is much lower than that of the pure CuBi_2_O_4_ and Ag_3_PO_4_, indicating that the combination of CuBi_2_O_4_ and Ag_3_PO_4_ has effectively improved the photogenerated carriers’ separation. Therefore, though a small amount of Ag^0^ has formed on the surface of CuBi_2_O_4_/Ag_3_PO_4_ (1:1), its photocatalytic performance is still enhanced compared with the pure CuBi_2_O_4_ and Ag_3_PO_4_.

Moreover, the free radical capture experiments were designed to investigate the difference of main reactive species in the photocatalysis systems of blank irradiation, CuBi_2_O_4_, Ag_3_PO_4_, and CuBi_2_O_4_/Ag_3_PO_4_ (1:1). Generally speaking, when the scavenger is added into the reaction system, if the degradation efficiency of the target contaminant do not change significantly, it indicates the weak effect of the captured reactive species on the photodegradation of the target pollutant, while if the degradation efficiency of the target contaminant is inhibited in a great degree, it indicates the captured reactive specie is involved in the photodegradation of the target pollutant and that the contribution of the reactive species is greater with the increased inhibition efficiency. Figure 8b–e shows the photodegradation efficiency of DS under various scavengers in different photocatalytic systems. From Figure 8b, O_2_^−^ is the major reactive species in the self-photodegradation process of DS, while the effect of *h*^+^ and OH can be ignored. In the photocatalytic system of the pure CuBi_2_O_4_, all the studied reactive species are involved in the degradation process of DS and the contribution order is *h*^+^ > O_2_^−^ > OH (see Figure 8c). The photodegradation efficiency of DS under different scavengers in the photocatalytic systems of Ag_3_PO_4_, as described in Figure 8d, indicates the important influence of OH and *h*^+^ on DS photodegradation. However, in the CuBi_2_O_4_/Ag_3_PO_4_ (1:1) composite photocatalytic system (see Figure 8e), O_2_^−^ and OH are the main reactive species for DS degradation and the influence of *h*^+^ is weak. Thus, there is a difference of the reactive species involved in the same pollutant degradation process under the different photocatalysis systems, finally leading to the differentiated photocatalytic mechanism of the catalysts.

On the basis of the results described above, the photolysis mechanisms of the blank irradiation, CuBi_2_O_4_, Ag_3_PO_4_, and CuBi_2_O_4_/Ag_3_PO_4_ reaction systems were proposed. In the blank irradiation system, DS should first absorb actinic photons and change to be DS *, then two processes may occur: (1) the direct photodegradation process of DS * and (2) the photooxidation process of DS; that is, the DS * reacts with the dissolved O_2_ to form O_2_^−^, which is subsequently involved in the degradation of DS. Because of the free radical capture experiment suggesting the O_2_^−^ has a significant effect on the DS degradation in the blank irradiation system, the photooxidation process of DS is the dominated process. This speculation has also been put forward in a previous paper [49], and the transition of photogenerated electrons and holes is described in Figure 9a. In the CuBi_2_O_4_ photocatalytic system, the electron–hole (*e*^−^–*h*^+^) pairs will generate under the visible light irradiation and, then, the *e*^−^ and *h*^+^ will induce to form other active radicals (such as O_2_^−^ and OH) during the separation/migration processes. Due to the oxidation potential of OH^−^/OH (1.99 eV) [27] being much higher than the potential of the valence band of CuBi_2_O_4_ (*E*_VB_ = 0.95 eV), the OH cannot be induced by the *h*^+^. However, the O_2_ absorbed on the surface of the catalyst can be induced to be O_2_^−^, owing to the lower position of the standard redox potential of O_2_/O_2_^−^ (0.13 eV) [50] than the potential of the conduction band (*E*_CB_ = −0.77 eV). Then, the partial O_2_^•−^ can be further induced to be OH (see Figure 9b). Therefore, the separated *h*^+^ and the induced O_2_^•−^/OH^•^ are involved in the degradation of DS. As for the photocatalytic system of Ag_3_PO_4_, the transition of photogenerated *e*^-^ and *h*^+^ are described in Figure 9c and the photogenerated *h*^+^ can be induced to form the active radical of OH^•^, whereas the photogenerated *e*^-^ could not be induced to form O_2_^•−^ because of the higher position of the standard redox potential of O_2_/O_2_^−^ than the potential of the conduction band (*E*_CB_ = 0.25 eV). Then, the separated *e*^-^ should be easily captured by the Ag^+^ coming from the lattice of the Ag_3_PO_4_, leading to the Ag^0^ deposition on the surface of catalyst. This phenomenon seriously influences the photocatalytic activity and stability of Ag_3_PO_4_. When the semiconductor of Ag_3_PO_4_ is combined with CuBi_2_O_4_ to form the composite photocatalyst of CuBi_2_O_4_/Ag_3_PO_4_, the formed Ag^0^ during the photocatalytic reaction process deposits on the surface of the catalyst and serves as the recombination center for the photogenerated *e*^-^ from CB of Ag_3_PO_4_ and the *h*^+^ from VB of CuBi_2_O_4_, resulting in the aggregation of the photogenerated *h*^+^ in the VB of Ag_3_PO_4_ and in the aggregation of the photogenerated *e*^-^ in the CB of CuBi_2_O_4_. Thus, the formation of O_2_^−^ induced by *e*^-^ from the CB of CuBi_2_O_4_ and the formation of OH induced by *h*^+^ from the VB of Ag_3_PO_4_ will be strong. From the free radical capture experiments, the effect of *h*^+^ on the photodegradation of DS is extremely weak, suggesting that *h*^+^ is mainly involved in the formation of OH. The detailed transition of photogenerated *e*^-^ and *h*^+^ in a photocatalytic system of CuBi_2_O_4_/Ag_3_PO_4_ is represented in Figure 9d, and its photocatalytic mechanism is consistent with the Z-Scheme theory [28].

## 4. Conclusions

Ag_3_PO_4_ was deposited on CuBi_2_O_4_ to construct Z-scheme photocatalyst, and it was used for DS degradation. CuBi_2_O_4_/Ag_3_PO_4_ exhibited an excellent photocatalytic performance under the system of visible-light irradiation. Free radical capture experiments suggest the active species of O_2_^−^ and OH contribute to the degradation of DS. Moreover, Z-Scheme theory can be used to explain the catalytic mechanism of CuBi_2_O_4_/Ag_3_PO_4_, and the formed Ag^0^ during the reaction process serves as the recombination center for the photogenerated *e*^-^ from CB of Ag_3_PO_4_ and the *h*^+^ from VB of CuBi_2_O_4_, which improved the separation of photogenerated carriers, finally leading to the enhancement of the catalytic performance.

## Figures and Tables

**Figure 1 nanomaterials-09-00959-f001:**
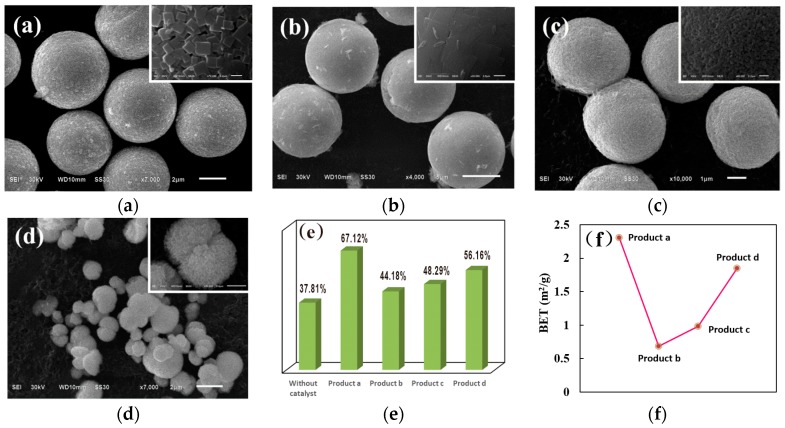
SEM images of CuBi_2_O_4_ obtained in different conditions: (**a**) 1.2 M NaOH and the step of “dilution” was after the addition of NaOH; (**b**) 1.2 M NaOH and the step of “dilution” was before the addition of NaOH; (**c**) 2.4 M NaOH and the step of “dilution” was after the addition of NaOH; and (**d**) 2.4 M NaOH and the step of “dilution” was before the addition of NaOH. (**e**) Photodegradation efficiency of diclofenac sodium (DS) in the catalytic systems of CuBi_2_O_4_ obtained in different conditions. (**f**) Specific surface area of different CuBi_2_O_4_.

**Figure 2 nanomaterials-09-00959-f002:**
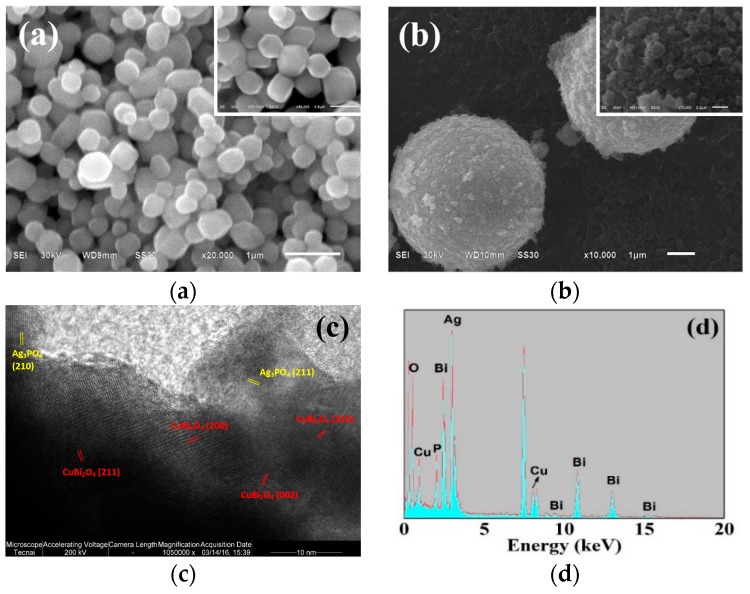
SEM images of (**a**) Ag_3_PO_4_ and (**b**) CuBi_2_O_4_/Ag_3_PO_4_ (1:1); (**c**) lattice fringe image; and (**d**) Energy-dispersive X-ray spectroscopy (EDS) pattern of CuBi_2_O_4_/Ag_3_PO_4_ (1:1).

**Figure 3 nanomaterials-09-00959-f003:**
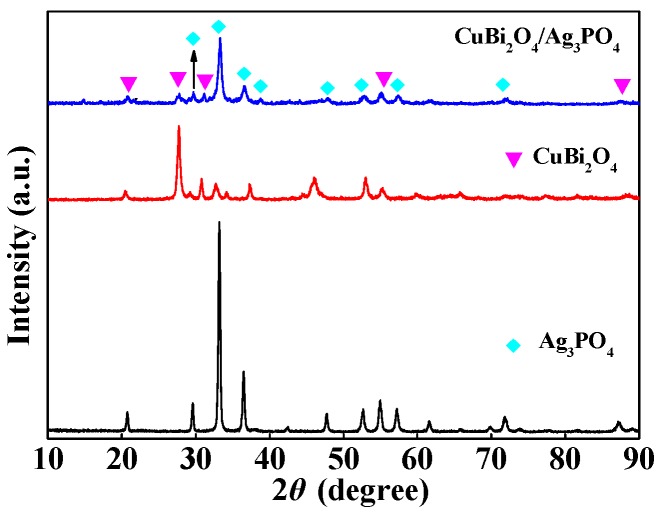
XRD patterns of Ag_3_PO_4_, CuBi_2_O_4_, and CuBi_2_O_4_/Ag_3_PO_4_ (1:1).

**Figure 4 nanomaterials-09-00959-f004:**
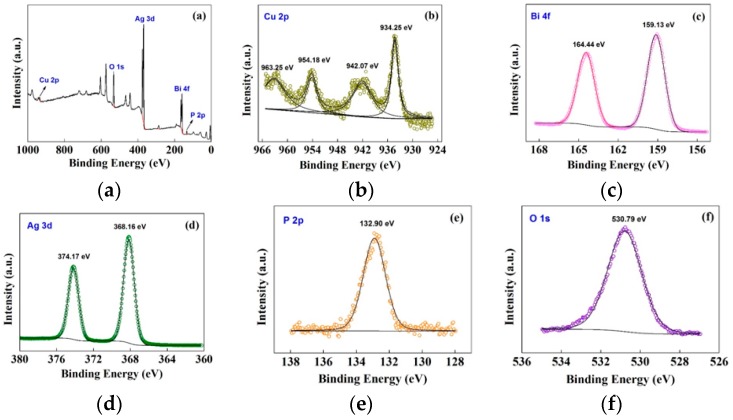
(**a**) XPS survey spectrum of CuBi_2_O_4_/Ag_3_PO_4_ (1:1) and high-resolution XPS spectra of (**b**) Cu 2p, (**c**) Bi 2p, (**d**) Ag 3d, (**e**) P 2p, and (**f**) O 1s.

**Figure 5 nanomaterials-09-00959-f005:**
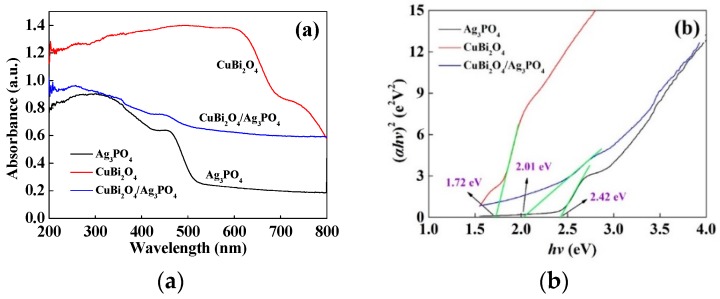
(**a**) UV-vis diffuse reflectance spectra and (**b**) plots of (*αhv*)^2^ versus energy (*hv*) of Ag_3_PO_4_, CuBi_2_O_4_, and CuBi_2_O_4_/Ag_3_PO_4_ (1:1).

**Figure 6 nanomaterials-09-00959-f006:**
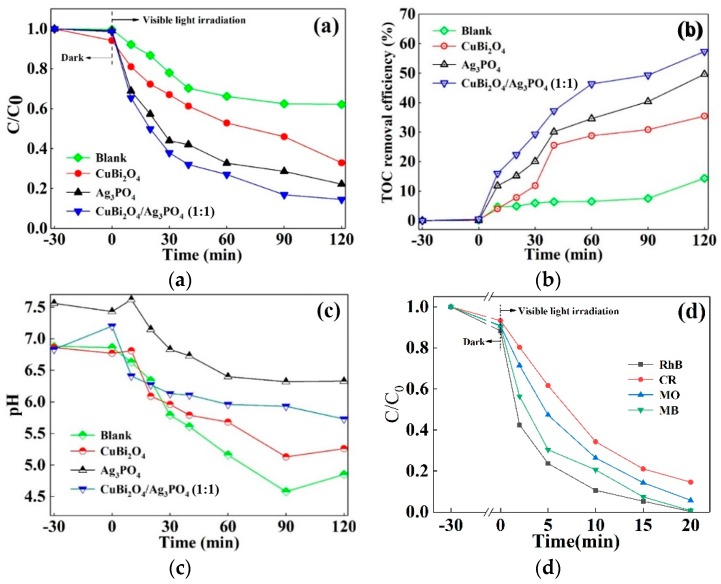
(**a**) Photocatalytic activity of the as-prepared catalysts toward DS degradation under visible light irradiation; (**b**) the TOC removal efficiency and (**c**) changes of pH values in the photodegradation solutions of different catalytic systems; and (**d**) the photocatalytic activity of CuBi_2_O_4_/Ag_3_PO_4_ (1:1) towards various dyestuff degradation under visible light irradiation.

**Figure 7 nanomaterials-09-00959-f007:**
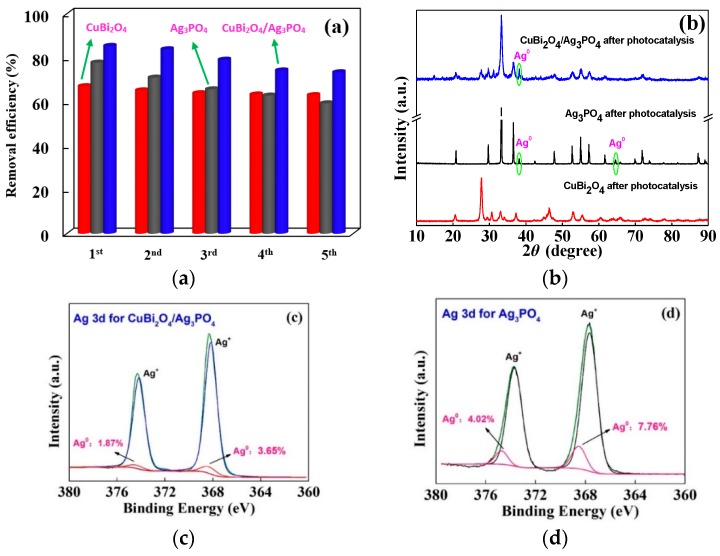
(**a**) Photodegradation efficiency of DS in different recycle runs with the various photocatalysts of CuBi_2_O_4_, Ag_3_PO_4_, and CuBi_2_O_4_/Ag_3_PO_4_ (1:1); (**b**) XRD pattern of the photocatalysts used for five recycling; (**c**) XPS spectrum of Ag 3d for the catalyst CuBi_2_O_4_/Ag_3_PO_4_ (1:1) recycled five times; and (**d**) XPS spectrum of Ag 3d for the catalyst Ag_3_PO_4_ recycled five times.

**Figure 8 nanomaterials-09-00959-f008:**
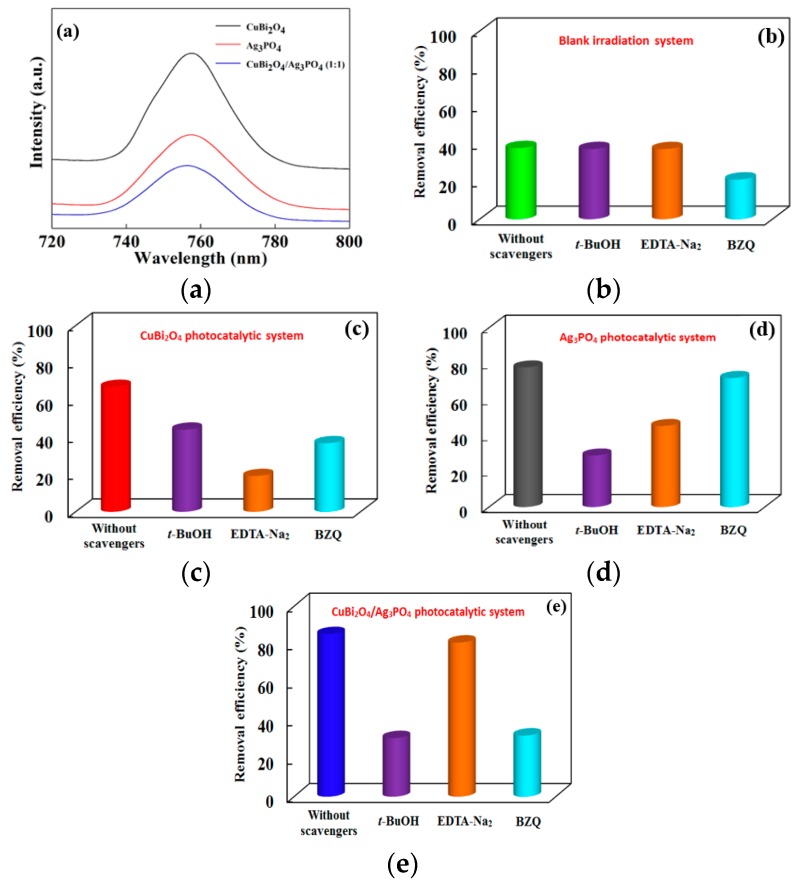
(**a**) Room temperature photoluminescence (PL) spectra of the as-prepared catalysts and photodegradation efficiency of DS under different scavengers in the photocatalytic systems of (**b**) blank irradiation, (**c**) CuBi_2_O_4_, (**d**) Ag_3_PO_4_, and (**e**) CuBi_2_O_4_/Ag_3_PO_4_ (1:1).

**Figure 9 nanomaterials-09-00959-f009:**
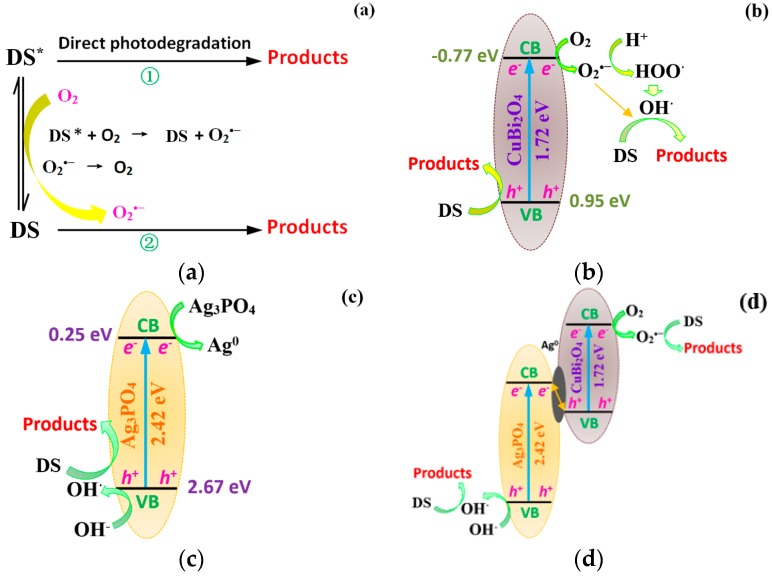
Transition of photogenerated electrons and holes in different photocatalytic system of (**a**) blank irradiation, (**b**) CuBi_2_O_4_, (**c**) Ag_3_PO_4_, and (**d**) CuBi_2_O_4_/Ag_3_PO_4_.

**Table 1 nanomaterials-09-00959-t001:** Photodegradation rate parameter of DS under different systems.

Catalyst	Specific Surface Area (m^2^/g)	*K*_app_ (min^−1^)	R^2^
Blank irradiation	/	0.0041	0.87
CuBi_2_O_4_	2.307	0.0084	0.95
Ag_3_PO_4_	3.874	0.0112	0.89
Commercial TiO_2_	/	0.0069	0.94
CuBi_2_O_4_/Ag_3_PO_4_ (2:1)	4.938	0.0098	0.92
CuBi_2_O_4_/Ag_3_PO_4_ (1:1)	7.846	0.0143	0.91
CuBi_2_O_4_/Ag_3_PO_4_ (1:2)	7.442	0.0138	0.91

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
