# Peer review of "Ag3PO4 Deposited on CuBi2O4 to Construct Z-Scheme Photocatalyst with Excellent Visible-Light Catalytic Performance Toward the Degradation of Diclofenac Sodium"

_nanomaterials, 2019, doi:10.3390/nano9070959_

Round 1
Reviewer 1 Report
Novel CuBi2O4/Ag3PO4 was prepared through the combination of hydrothermal treatment and in-situ deposition method and their textures were characterized by XRD, XPS, SEM/HRTEM, EDS, UV-Vis and PL. The catalytic degradation of diclofenac sodium (DS) of the prepared CuBi2O4/Ag3PO4, CuBi2O4 and Ag3PO4 was evaluated. Furthermore, the degradation mechanism of the prepared catalysts was proposed through scavenger test and PL analysis. Overall the text was written well. However, some issues must be declared.
Abstract: Novel CuBi2O4/Ag3PO4 was prepared…and PL. “Novel” means no research or idea was proposed such nanocomposites before. Please check it as possible you can. Applied Catalysis B: Environmental, 2017, 209, 720-728. Materials, 2018, 11(4), 511.
The degradation efficiency and mineralization rate of DS by using the prepared CuBi2O4/Ag3PO4 were 85.45% and 57.33%, which are similar to those of CuBi2O4 and Ag3PO4. What is the major advantages of the prepared CuBi2O4/Ag3PO4 comparing to CuBi2O4 and Ag3PO4? What is the reason we want to use CuBi2O4/Ag3PO4 to replace CuBi2O4 and Ag3PO4 catalysts?
If possible, please evaluate the DS degradation efficiency comparing to N-doped TiO2 or commercial TiO2 catalysts.
How about practicality of the prepared CuBi2O4/Ag3PO4 for the degradation various dyestuff in environmental water samples under sunlight irradiation.
Author Response
Comment 1: Abstract: Novel CuBi2O4/Ag3PO4 was prepared…and PL. “Novel” means no research or idea was proposed such nanocomposites before. Please check it as possible you can. Applied Catalysis B: Environmental, 2017, 209, 720-728. Materials, 2018, 11(4), 511.
Response 1:Thanks for your careful comment and suggestion. The word “Novel” has been deleted from the Abstract.
Comment 2: The degradation efficiency and mineralization rate of DS by using the prepared CuBi2O4/Ag3PO4 were 85.45% and 57.33%, which are similar to those of CuBi2O4 and Ag3PO4. What is the major advantages of the prepared CuBi2O4/Ag3PO4 comparing to CuBi2O4 and Ag3PO4? What is the reason we want to use CuBi2O4/Ag3PO4 to replace CuBi2O4 and Ag3PO4 catalysts?
Response 2:Compared to CuBi2O4 and Ag3PO4, the major advantages of the prepared CuBi2O4/Ag3PO4 is that both of the structure stability and catalytic stability enhances in a large extent. Moreover, in order to obtain the same photodegradation efficiency of DS, the catalyst dosage of Ag3PO4 in the composite CuBi2O4/Ag3PO4 is much less than that of the pure Ag3PO4.
Comment 3: If possible, please evaluate the DS degradation efficiency comparing to N-doped TiO2 or commercial TiO2 catalysts.
Response 3:Thanks for your kind advice. The photocatalytic efficiency of commercial TiO2 has been carried out. Please pay attention to “Table 1” and “page 7-8, line of 260-272”.
Comment 4: How about practicality of the prepared CuBi2O4/Ag3PO4 for the degradation various dyestuff in environmental water samples under sunlight irradiation.
Response 4:Thanks for your careful comment and suggestion. The photocatalytic activity of CuBi2O4/Ag3PO4 toward various dyestuff had been operated. Please pay attention to “Figure 6(d)” and “page 9, line of 288-292”.
Reviewer 2 Report
This paper did systematic research on the Z-scheme photocatalyst: Ag3PO4 deposited on CuBi2O4. Authors discussed the properties of Ag3PO4/CuBi2O4 system and its photocatalytic activity in relation to diclofenac sodium degradation. The valuable part of the manuscript is the discussion about the photocatalytic mechanism. The results are significant. The manuscript can be published after the following minor revisions:
1. Authors provided photocatalytic experiments under visible light irradiation but there is no information about the range of visible light irradiation. Was any cut-off filter used in the experiments?
2. What is the reason that CuBi2O4 represented as “product a” showed highest photocatalytic activity? What is the role of the particle morphology? Was BET specific surface area measured for the samples a-d?
3. The mass ratio between Ag3PO4 and CuBi2O4 was 1:1. Why? Why not 1:2 or 2:1?
Author Response
Comment 1: Authors provided photocatalytic experiments under visible light irradiation but there is no information about the range of visible light irradiation. Was any cut-off filter used in the experiments?
Response 1:Thanks for your careful comment. The rigorous sentence of “equipped with a 300 W Xe lamp and an ultraviolet cutoff filter of providing visible light ≥ 400 nm” had been displayed in the revised manuscript. Please pay attention to “page 3, line of 109-110”.
Comment 2: What is the reason that CuBi2O4 represented as “product a” showed highest photocatalytic activity? What is the role of the particle morphology? Was BET specific surface area measured for the samples a-d?
Response 2:BET specific surface area of the samples had been measured, and the detailed explanation can be seen from “Figure 1(f)” and “page 5, line of 175-178”.
Comment 3: The mass ratio between Ag3PO4 and CuBi2O4 was 1:1. Why? Why not 1:2 or 2:1?
Response 3:Thanks for your kind advice. The mass ratio of 1:2 and 2:1 had been supplemented. Please pay attention to “Table 1” and “page 7-8, line of 260-272”.
Round 2
Reviewer 1 Report
All questions are provided good responses. It can be published in present form.